# Second Victims in Intensive Care—Emotional Stress and Traumatization of Intensive Care Nurses in Western Austria after Adverse Events during the Treatment of Patients

**DOI:** 10.3390/ijerph19063611

**Published:** 2022-03-18

**Authors:** Samuel Ganahl, Mario Knaus, Isabell Wiesenhuetter, Victoria Klemm, Eva M. Jabinger, Reinhard Strametz

**Affiliations:** 1Health University of Applied Sciences Tyrol, Innrain 98, 6020 Innsbruck, Austria; samuel.ganahl@hotmail.com (S.G.); mario.knaus@gmx.at (M.K.); eva.jabinger@fhg-tirol.ac.at (E.M.J.); 2Vorarlberger Landeskrankenhäuser, Interdisziplinäre Intensivstation, 6800 Feldkirch, Austria; 3Paediatric Intensiv Care Unit, Tirol Kliniken GmbH, 6020 Innsbruck, Austria; 4Wiesbaden Business School, RheinMain University of Applied Sciences, 65183 Wiesbaden, Germany; i.wiesenhuetter@gmail.com (I.W.); victoria.klemm@student.hs-rm.de (V.K.); 5Munich University Institute for Psychological Psychotherapy Training (MUNIP), 80802 Munich, Germany; 6Head of Wiesbaden Institute for Healthcare Economics and Patient Safety, RheinMain University of Applied Sciences, 65183 Wiesbaden, Germany

**Keywords:** second victim, intensive care, risk management, support programs, coping strategies, patient safety

## Abstract

Background: The second victim phenomenon is common among nurses in intensive care units. Apart from quantitative studies, little is known about individual cases among those high-risk groups. This study evaluates the natural history and cause of second victim traumatization in Western Austria for the first time to tailor specific intervention. Methods: A total of 20 guided interviews were conducted with intensive care nurses in Western Austria. All interviews were transcribed and analyzed with MAXQDA software. Evaluation followed the structuring qualitative content analysis scheme according to Kuckartz. Results: The most frequent symptoms of the second victim phenomenon reported were feelings of guilt and problems with falling asleep. Coping with the second victim phenomenon was accomplished by conversations in private as well as among work colleagues. Conclusions: Intensive care nurses are exposed to many exceptional situations which have a high likelihood of resulting in second victim traumatization. As proximal psychosocial support is considered to be a main source of coping, wide-spread implementation of effective psychosocial peer support programs ought to be applied by medical organizations. Patient safety measures such as proactive and reactive clinical risk management (e.g., CIRS) should be linked to second victim support.

## 1. Introduction

The term second victim was first coined in 2000 by Professor Albert Wu [1]. A second victim was associated with emotional distress due to treatment errors or incidents. In 2009, the term was expanded and defined in more detail to include people who work in the medical field and who become victims due to an unforeseeable incident during the treatment of a patient, a medical error, or an injury to a patient. The traumatization caused by this event makes the treating person become a second victim [2].

Due to the prevailing COVID-19 pandemic, the associated exceptional situation in health institutions can be viewed as an unforeseen incident, and the resulting traumatization and stress can lead to treating practitioners becoming second victims [3,4]. In addition, staff are overloaded and stressed in every aspect due to the scarcity of resources in the health care system, resulting in an increased risk for medical errors [5]. 

In the Anglo-American region, second victim phenomenon in inpatient acute care has been well researched [6,7] and is also familiar to staff. Evidence indicates that around 10–42% of treating physicians have already experienced second victim phenomenon in their professional careers [6]. Among young German physicians in internal medicine, 59% experienced second victim incidents in their career so far [8] and among German nurses, while 60% experienced the second victim phenomenon at least once in their working lifetime [9]. 

Second victim traumatization may have negative impacts on health worker wellbeing due to dysfunctional coping strategies, such as self-isolation, reduced self-confidence, flash backs, and increased fear of making mistakes [7]. As psychological precursors are a source of increased risk for medical errors [10], they are also likely to affect the safety of future patients treated after second victim traumatization.

In comparison to the Anglo-American region [11,12], no systematic tailor-made psychosocial peer support programs have been established in German-speaking countries so far. In order to obtain a deeper understanding of the natural course of second victim traumatization and the demand for support in high-risk populations, such as intensive care nurses, this explorative qualitative study was conducted in the Western Austrian regions of Vorarlberg and Tyrol.

The following research questions were defined based on previous research in other regions of the world [6,13,14,15] in order to detect similarities and differences in settings in German-speaking areas:Which subjective physical and psychological effects do traumatic events in everyday working life have on intensive care nurses in Western Austria?What coping strategies have they developed to cope with their second victim traumatization?Which additional measures should be implemented to minimize the development of second victim phenomenon?To what extent does COVID-19 play a role for second victim traumatization?

## 2. Materials and Methods

To answer these research questions, a qualitative research approach was used, in which small random samples in Western Austria were asked to respond in detail to open questions on many aspects of the topic of “Second Victim in Intensive Care” [16]. 

A semi-structured interview guide was developed on the basis of the content-related topics and research questions by combining trial interviews with literature research on the topic of second victim. Independent cognitive trial interviews were first carried out with two intensive care nurses who had already experienced second victim phenomenon to gain insight into the cognitive processes of the interviewees responding to the questions. 

The final interview guide is divided into different phases. In the introductory phase, the study information, the relevance of the topic, and the declaration of consent were explained. All participants were given information on second victim phenomenon to ensure everyone had the same level of knowledge on the subject and to avoid misunderstandings. In addition, participants were asked whether they had already experienced second victim phenomenon in the sense of the given definition. In the narration phase, the research participants were asked to report on what they had experienced. After the three main deductive categories could be formed based on the SPSS principle in connection with the research questions—i.e., stress, experience of support measures (coping strategies), and precautions that could have been taken to avoid development of second victim phenomenon—additional questions were created under the respective categories to ensure that all topics were covered by answering the research questions. 

A total of 20 intensive care nurses from Tyrol and Vorarlberg were interviewed. Sample size was set to this limit to address expected redundancy of information. To find participants, the research project was put out for tender and distributed to the intensive care units in Western Austria (sampling through self-activation). Intensive care nurses could then contact us by phone or email if they had already suffered from emotional stress or trauma after an adverse event in an intensive care unit and were willing to be questioned. The snowball principle was also applied, in which respondents were asked about other possible interview partners [17]. 

Ten intensive care nurses that suffered second victim phenomenon in Vorarlberg and ten in Tyrol were interviewed. By surveying twenty participants in total the research team gained an overview of the selected topic. In addition to the inclusion criteria, care was taken to include intensive care nurses from different locations in Vorarlberg as well as in Tyrol in order to assess traumatization in different health organizations [16]. The gender distribution was stratified to 80% women and 20% men, corresponding to gender distribution of intensive care nurses in Vorarlberg and Tyrol [18]. The location of the interview was determined at the request of the research participants. Due to the prevailing pandemic, four interviews were carried out online using the Zoom Meetings app. The remaining 16 interviews were carried out at the study participants’ homes. All interviews were conducted between participant and a member of the research team only (SG or MK).

Respondents described and recorded experiences from their everyday professional lives in the sense of phenomenology. Intensive caregivers described their subjective feelings in these situations as well as their experiences in detail [16]. 

The interviews were evaluated using the MAXQDA software, and the individual interviews were cited according to the lines generated by the software.

## 3. Results

A total of 20 intensive care nurses in Tyrol and Vorarlberg were interviewed. Characteristics of study participants are shown in Table 1. 

### 3.1. Symptoms

In this study, various psychological and physical symptoms, as well as effects on everyday working life, were identified. Table 2 illustrates how often participants reported these symptoms because of an adverse event.

The most common symptoms of respondents were feelings of guilt, problems falling or staying asleep, flashbacks of what they had experienced, and problems in everyday work (such as reduced performance and problems with performing routine activities). Self-doubt also contributed to said problems. In their private lives, the participants experienced a reduction in life quality and internal unrest, such as feeling queasy. The following quotes serve as examples of how participants experienced the symptoms:


*“I felt guilty, I could barely fall asleep at night because it really affected me greatly”, “… I still had a relatively great feeling of guilt. Again and again I thought about it…”, “… at times I was aggressive towards myself because I was the one who made the mistake”, “I doubted and asked myself if there was anything else I could have done”, “… I must say that ever since I’m very gloomy and think about it a lot. I wouldn’t call it depressions though, more like a bad feeling and sadness that it happened even though the patient survived”, “yes it’s like a constant circling. I couldn’t help but think about it”, “back then I had problems sleeping and I still don’t sleep well and am actually exhausted. Yes.”, “I’d say sleep loss did affect my ability to work.”, “I was definitely off track and it took a while for me to get back into my work routine.”, “well until the end, the end of my apprenticeship I didn’t like going to that place and that was the case for over a year. So, you can say I avoided it for more than a year.”*


### 3.2. Coping Strategies and Risk Management

Participant experience of support measures is illustrated in Table 3. 

The participants surveyed mentioned conversations both in their private surroundings and with colleagues, as well as exercising in their free time as the most common coping strategies to clear their heads. Three people in this survey stated that they had taken advantage of professional support to be able to deal with the second victim phenomenon. Ten people explained that they had changed their work routines to process what they had experienced. The following quotes serve as examples of the coping strategies of the interviewees:


*“I’m still afraid that if unplanned recordings come out of the shock room (…) then I’ll get backup so that they can take over for me if necessary”, “yes, talk to friends or acquaintances. About such a similar topic, etc.”, “In the acute situation especially, to talk to my colleagues about it again and (…) to reflect”, “Over time it was no longer so relevant. It took time.”, “Going out into the fresh air, doing sport, just (…) going into nature to process it.”, “We have a psychologist in-house, with whom I talked and worked a lot.”*


Reactive risk management regarding patient safety and the avoidance of the second victim phenomenon in the respective intensive care unit was divided into the following categories: critical incident reporting system (CIRS), supervision, and case discussions, based on the 20 interviews. Participants stated that supervision and case discussions were implemented in their day-to-day work in their intensive care unit. Offers for supervision and case discussions were increasingly used in the context of the COVID-19 pandemic. In addition, the desire for regularity and interdisciplinarity was expressed. The following quotes describe the reactive risk management interviewees have experienced so far:


*“Well, I know that we just started doing case discussions but I must admit, I’ve never had such a case discussion to this point”, “Since Corona supervisions are offered…”, “maybe supervisions should be mandatory after an adverse event. It certainly can’t do any harm”, “yes, well, CIRS reports are written but I feel like when I write a report, sometimes something happens and then again nothing.”*


Regarding preventive measures as a part of risk management to avoid adverse events, and thus second victim phenomenon, the respondents mentioned checklists, the four-eyes principle, training, and care rounds. Table 3 shows how many interview participants considered the measures to be fundamental to increase patient safety and to avert the potential of second victim phenomenon occurring. The participants wanted such measures to be taken more regularly and to be better integrated into the daily routine which the following quotes exemplify:


*“care rounds can actually mitigate these things, I really do believe that.” “Maybe more training, like simulation training. I’ve worked for seven years and maybe participated in three simulations. That’s not enough.”*


## 4. Discussion

Only a few of the participants had heard the term before participating in this research. The most common symptoms reported by participants were feelings of guilt, problems falling or staying asleep, flashbacks, and problems in everyday work. The participants reported conversations and exercising as the most common coping strategies. Participants stated that reactive risk management had been implemented in their day-to-day work in their intensive care unit. In addition, the desire for regularity and interdisciplinarity was expressed. Preventive measures such as nursing rounds, the four-eyes principle, checklists, and training are endorsed in order to improve the safety culture in such a high-tech area as intensive care. 

Patient safety may also be improved by the use and encouragement of CIRS. As far as the training and implementation of CIRS in hospitals is concerned, according to the statements of the intensive care nurses of this study, there is still a need for action to create the basic framework conditions for a CIRS. So far, the focus lies on patient safety; thus, the perspective here should be broadened with consideration for the second victim phenomenon. The measures in the context of risk management could improve patient safety and prevent the development of the second victim phenomenon in intensive care staff [19]. Furthermore, the fact that there are no contact points for second victims in German-speaking countries should not be disregarded. There are support measures such as CIRS in the hospitals; however, they are not specifically aimed towards second victims. A recommendation should be given to practitioners to reconsider or implement the support offer. The benefits of various programs to support second victims also have a positive effect on the costs of the health care system and the health of employees [20]. This justifies the broad implementation of effective support programs in Austria. The COVID-19 pandemic, and its unpredictable consequences for the mental health of medical staff, also underlines the relevance of offers, especially for second victims [5]. 

According to the literature, the error rate in the hospital sector is between 4% and 18%. Most errors occur with pharmaceuticals (e.g., through mix-ups), when using medical products and in teamwork (e.g., communication errors) [21,22]. This is confirmed by the results of the interviews conducted in this study. The development of second victim phenomenon due to the COVID-19 pandemic could represent an interesting starting point for further research. If this pandemic is compared with the SARS pandemic of 2002–2003, it can be assumed that at least half of all those treating patients must struggle with emotional stress or trauma and could develop second victim phenomenon as a result [13]. 

Regarding symptoms, the literature shows that around two thirds of the respondents process the event they experienced in a dysfunctional manner. Signs of this include sleep disorders, feelings of guilt, isolation, depression, or reliving the situation (flashbacks) [23]. The statements made by those affected in the interviews are reflected in the literature. These symptoms can be compared internationally with the results of Harrison et al. [24] and Waterman et al. [13]. These studies on doctors across Great Britain, the USA, and Canada also showed sleep disorders, stress, anxiety in general, and future mistakes as the most common symptoms of second victims [23,24]. Regarding the recovery of second victims, Gazoni et al. [25] indicate that around 10% of all those affected will not recover from the event. This coincides with the results of the interviews: 2 of the 20 respondents stated that they had not fully recovered from the event they had experienced, although the experiences of the intensive care staff concerned were already two and five years ago. It should be noted that the consequences and effects can vary from person to person. According to Grissinger [26], such effects can range from post-traumatic stress disorders to giving up a job or, in the worst case, suicide. One of the respondents had transferred to a different unit, but none of the respondents had suicidal thoughts. In the study by Gazoni et al. [25], approximately 12% of the respondents transferred jobs, which is comparable with the data in this survey. In the present survey of intensive care nurses in Western Austria, around half of the second victims surveyed stated that they had thoughts of resigning or transferring jobs. This can probably be justified with the situation in the intensive care units due to the COVID-19 pandemic. According to the German Society for Internal Intensive Care and Emergency Medicine (DGIIN) (2021), 72% of non-medical staff in intensive care units said they felt a feeling of overload in the wake of the pandemic, and a third of them said they would want to transfer jobs within the next twelve months. According to the study by Gazoni et al. [25], 62% of the participants surveyed feared legal consequences. In the present study, 40% of those questioned feared legal consequences. This result roughly coincides with international studies. A deviation cannot be ruled out, however, since most of the studies were carried out with doctors and not with intensive care nurses. Such differences leave room for interpretation for future research in the field of intensive care in relation to second victim phenomena. The literature describes an increased susceptibility to errors by second victims with their patients in the future. The interviews with the intensive care nurses in Western Austria show that those affected were more focused in their work and paid more attention to ensuring that no further mistakes occur. If second victims care for patients in the future, these patients will also suffer consequences, since reduced performance and problems in everyday work can lead to an increased susceptibility to errors [27]. According to Harrison et al. [24], 81% of those surveyed are afraid of future mistakes caused by flashbacks that remind them of what they have experienced in their everyday work. In this study, 17 study participants indicated that they either suffered from an increased need to double-check their work and themselves or from flashbacks, as the event had a strong impact on their way of thinking. Internationally, there is presumably a comparability of the results; however, this would have to be verified by further research. 

Gazoni et al. [25] stated that 75% of the anesthesiologists surveyed indicated that talking to colleagues was a measure of coping with emotional stress or trauma. If the collegial help within the department is insufficient, the employer should offer crisis intervention by a special team [2]. This also includes debriefings, which, however, were not used by the intensive care nurses in this survey in most cases. This could be because case discussions and supervision by employers were largely only introduced since the high level of exposure during the COVID-19 pandemic, and the respondents had not previously taken part in measures of this kind. If these measures are no longer sufficient, professional support should be considered [2]. According to Edrees et al. [12], around half of second victims only seek help after a few days when they realize they have been traumatized by the experience; therefore, close monitoring of second victims should not be neglected. 

Regular simulation training is recommended in international comparative studies to increase patient safety [24,25]. To improve patient safety and prevent the second victim phenomenon among employees, these measures are an important part of everyday ward life [5,28]. The respondents in this study were also asked about support offers that were currently available to them. The survey and the literature show that family and friends play a major role. It makes little difference whether family members or friends work in a medical environment or not [29]. In the literature, it is mentioned that in the Anglo-American area more and more programs are implemented directly in health care facilities to support second victims. These are, for example, the RISE program of the Johns Hopkins University or the forYOU program of the University of Missouri [11,26]. Through these programs, positive medical effects could be achieved and costs could be saved. There are only a few voluntary initiatives in German-speaking countries. In the German-speaking region, there is no nationwide support from any of the programs [5,30]. These experiences also apply to the present study of intensive care workers in Western Austria. The respondents stated that various measures and contact points were implemented, but that they were mostly not used. On the one hand, this is because these measures were only made possible at relatively short notice in the wake of the COVID-19 crisis, and on the other hand, because they were mostly covered by their superiors. The reluctance to discuss this topic with superiors is reflected in both the interviews and the literature. Ullström et al. [29] state that support from superiors is not always satisfactory and can even have a further disappointing effect on second victims. Peer groups could be a good alternative, as empathy and understanding that is shown to a person from within one’s own ranks is perceived as an immensely great support. Regarding the three-step model of supporting second victims according to Scott et al. [2], if the collegial help within the department is not sufficiently effective, then crisis intervention by a special team or even a network of professional support should be considered. To improve the CIRS application and thereby create a better safety culture, an increased need for training is necessary at all levels of hospital management [31]. Additional reactive measures to avoid the second victim phenomenon are: short breaks, offering collegial discussions, debriefing of stressful situations and technical support, carrying out an error analysis, and avoiding blame and bullying in the team [5]. The possibility of a break after an event was desired in the interviews but was not given due to time and personal reasons. The fear of exclusion from the team also played a central role in the interviews and should be dealt with together with the subject of blaming and bullying to achieve a better culture of discussion and safety in the workplace. Active collegial conversation in the workplace strengthens the self-esteem of those affected, provides support from within the team, and makes a significant contribution to coping with the second victim phenomenon. As underpinned by the literature, error analyses in the context of debriefings in the interdisciplinary team were named as suggestions for improvement. An open culture of conversation can help prevent thesecond victim phenomenon, and clear communication helps those affected to overcome the difficult time. In addition, strengthening the resilience of employees can be helpful if the staff reaches their limit, as the COVID-19 pandemic currently shows [5].

Regarding the limitations, it should be mentioned that the selection of the interview partners attempted to obtain the broadest possible overview of the topic of “Second Victims in Intensive Care in Western Austria”. In the 20 interviews carried out, however, not all subject areas on second victims could be recorded and processed. Furthermore, limitations of the participants and centers involved may limit the representability. Nevertheless, an attempt was made to obtain different opinions and views from the respondents, and despite the small number of interviews, high-quality data were generated on certain subject areas. The theoretical background on which the question was based mainly relates to studies from the Anglo-American region, as German-language literature hardly exists, which is also to be regarded as a limitation. It should also be considered that many of the studies that have already been carried out relate to medical personnel. The possibility of international comparison regarding intensive care nurses is, therefore, limited. However, since a large proportion of the medical staff is employed in nursing, more attention should be paid to them in the future. Regarding risk management, it must be noted that there are different risk management systems and different safety culture settings in different intensive care units with different categories. It is, therefore, difficult to compare the various intensive care units. Another limitation is that some respondents may not want to tell the researchers everything because of the role change problem and the fear that their statements could have an impact on their workplace. 

## 5. Conclusions

The second victim phenomenon is present among intensive care nurses in Western Austria and, therefore, requires further research. As expressed, potential coping strategies for the phenomenon can be compared with previous research in different settings, as well as with further quantitative surveys of intensive care nurses, and tailored intervention programs addressing prevention and systematic support of second victims should be considered for this target group. 

## Figures and Tables

**Table 1 ijerph-19-03611-t001:** Characteristics of study participants.

Characteristic		Number of Participants
work experience in intensive care	1–5 years	4
	6–10 years	10
	11–15 years	3
	16 years and above	3
familiarity with the term second victim	yes	5
	no	15
type of event which caused symptoms	drug related	9
	medical device related	3
	interaction with relatives	4
	intubation problems	2
	COVID-19 pandemic	2

**Table 2 ijerph-19-03611-t002:** Symptoms after an adverse event.

Symptoms		Number of Participants
psychological symptoms	feelings of guilt	20
	anxiety	12
	burnout	2
	depression	4
	internal unrest	12
	drop of life quality	9
	self-doubt	4
	aggression	3
physical symptoms	sweating	3
	palpitation	1
	racing heart	2
	crying	4
	insomnia	5
	nausea	1
	difficulty sleeping	10
	fatigue	1
effects on everyday working life	decreased efficiency	6
	problems with the work routine	5
	increased controlling	11
	increased errors	2
	flashbacks	11

**Table 3 ijerph-19-03611-t003:** Support measures to minimize symptoms of second victim phenomenon.

Support Measures		Number of Participants
coping strategies	work processes changed/rituals	10
	private conversations	11
	conversation with colleagues	11
	time to cope	3
	sports	12
	professional support	3
reactive risk management	supervision	12
	case discussions	12
	CIRS	20
preventive risk management	nursing rounds	7
	checklists	14
	four-eyes principle	13
	trainings	14

## Data Availability

The data presented in this study are available on request from the corresponding author.

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
