# Peer review of "Second Victims in Intensive Care—Emotional Stress and Traumatization of Intensive Care Nurses in Western Austria after Adverse Events during the Treatment of Patients"

_ijerph, 2022, doi:10.3390/ijerph19063611_

Round 1
Reviewer 1 Report
The review report is attached
Kind regards

Author Response
Dear Reviewer,
First of all thank you very much for your valuable comments that helped us to improve our paper. We would like to adress each of your comments point by point as followed.
Kind regards
Reinhard Strametz on behalf of all authors
INTRODUCTION
• →Due to the prevailing COVID19 pandemic, the associated exceptional situation in health institutions can be viewed as an unforeseen incident, and the resulting traumatization and stress can lead to the treating practitioners becoming second victims [3, 4]. The staff is additionally overloaded and stressed in every aspect due to the scarcity of resources in the health care system [5]. ← ..... Argue this paragraph. The previous paragraph mentions that the second victims are related to a medical error or injury of the patient. Work stress has negative consequences for health workers, but it is due to the workload. Stress must be
related to medical errors. With this contribution, it connects with the previous paragraph and with research.
This is a very valid point. We have emphasized the link between stress and medical errors. This section now reads:
Due to the prevailing COVID19 pandemic, the associated exceptional situation in health institutions can be viewed as an unforeseen incident, and the resulting traumatization and stress can lead to the treating practitioners becoming second victims [3, 4]. In addition staff is overloaded and stressed in every aspect due to the scarcity of resources in the health care system resulting in an increased risk for medical errors [5].
• →The following research questions were defined based on previous research in other regions of the world in order to detect similarities and differences in settings in German speaking area← ..... Bibliographic references.
We have added the following references in the text:
- Scott, S. D.; Hirschinger, L.; Cox, K.; McCoig, M.; Hahn-Cover, K. E.; Epperly, K. M.; Phillips, E.; Hall, L. Caring for Our Own: Deploying a Systemwide Second Victim Rapid Response Team. Comm. J. Qual. Patient Saf. 2010, 36 (5), 233–240.
- Edrees, H; Connors, C.; Paine, L.; Novell, M.; Taylor, H.; Wu, A. W. Implementing the RISE second victim support programme at the Johns Hopkins Hospital: a case study. BMJ 2016, 6, 1–12.
- Watermann, A. D.; Garbutt, J.; Hazel, E.; Dunagan, W. C.; Levinson, W.; Fraser, V. J.; Gallagher, T. H. The Emotional Impact of Medical Errors on Practicing Physicians in the United States and Canada. Comm. J. Qual. Patient Saf. 2007, 33, 467–476.
- Mira, J. J.; Lorenzo, S.; Carrillo, I.; Ferrús, L.; Pérez-Pérez, P.; Iglesias, F.; Silvestre; c.; Olivera, G.; Zavala, E.; Nuno-Solinis, R.; Maderuelo-Fernández, J. A.; Vitaller, J.; Astier, P. Interventions in health organisations to reduce the impact of adverse events in second and third victims. BMC Health Services Research 2015, 15, 341
- Moreno-Mulet, C.; Sansó, N.; Carrero-Planells, A.; López-Deflory, C.; Galiana, L.; García-Pazo, P.; Borràs-Mateu, M. M.; Miró-Bonet, M. The Impact of the COVID-19 Pandemic on ICU Healthcare Professionals: A Mixed Methods Study. J. Environ. Res. Public Health 2021, 18, 9243
MATERIALS AND METHODS
• Does this research have a code of approval from an ethics committee?
We clarified this in the section below conclusion as followed:
Institutional Review Board Statement: Ethical review and approval were waived for this study, due to anonymous and voluntary participation in this survey.
• →Due to the prevailing pandemic, four interviews were carried out online←.... Detail how these interviews were conducted (means used, for example zoom or moodle...)
We added the missing information. It now reads:
Due to the prevailing pandemic, four interviews were carried out online using zoom meeting app.
DISCUSSION
• In the discussion of results there is too much information referring only to results. The first part of this section must be restructured. Transfer to the results part everything that will not be discussed. Only a summary of the results to be discussed should be included in the results discussion.
We have shortened this section.
• →The measures in the context of risk management could improve patient safety and prevent the development of a second victim phenomenon in intensive care staf←.... Bibliographic references.
We have added a reference
• Include more limitations. There is a very small sample. With more interviews and more centres, the study would have been more representative.
We have rephrased this section. It now reads:
Regarding the limitations, it should be mentioned that the selection of the interview partners attempted to obtain the broadest possible overview of the topic of “Second Victims in Intensive Care in Western Austria”. In the 20 interviews carried out, however, not all subject areas on second victims could be recorded and processed. Also limitation of participants and centers involved may limit representability.
Reviewer 2 Report
The manuscript ijerph-1621457 entitled "Second Victims in Intensive Care- Emotional stress and traumatization of intensive care nurses in Western Austria after adverse events during the treatment of patients" is a phenomenological study that analyses the experience of nurses in intensive care units in Western Austria who are considered second victims. It aims to answer the cause of traumatisation, following an adverse event, and the subsequent negative effects on the professionals, by analysing the coping strategies employed by the professionals in order to suggest/adapt a specific intervention to reduce and facilitate coping/overcoming.
The manuscript presented is clear, concise and structured. The research question is explicitly stated and well delimited. The design, methods and instruments used are appropriate and well justified. The qualitative content analysis is appropriate and the results are presented in an orderly manner, answering the research questions. The use of tables is appropriate and well-adjusted. The discussion is presented in an orderly and well-integrated manner. However, the conclusions are vague.
For these reasons, a minor revision of the present version is recommended before publication.
Specific remarks:
1- Introduction:
Although the concept of second victim is clearly defined, it encompasses several characteristics. It is suggested to include the specific characteristics or criteria used by the researchers to consider the nursing professional as a second victim and therefore a suitable subject as an informant.
I also consider it useful to include in the introduction the concept of patient safety mentioned later in the discussion.
2. Methods:
The methodological design is appropriate and conforms to international COREQ recommendations. The approach, as a phenomenological study, responds to the object of study. The techniques used in the data collection are in accordance with the type of design and are clearly described, as well as the sampling technique (snowball).
As a suggestion for improvement, the question arises as to why there are only 20 individuals, and supposing that it responds to criteria of redundancy of information, it is suggested that this question be explicitly reflected in the manuscript.
3. Results:
Table 2 refers to Symptoms after an adverse event. It reflects psychological and physical symptoms such as internal unrest and drop of quality of life. These symptoms can be interpreted in different ways and are ambiguous, so it is suggested to include the meaning used in the text.
4. Discussion:
Delete the opening sentence : A total of 20 interviews took place in Tyrol and Vorarlberg. It is already included in the results section.
5. Conclusion:
The conclusions are vague and do not give a clear description of the meaning of the phenomenon studied. The phenomenological design selected in the study does not allow, nor is it intended to allow, for generalisation. The sentence "The second victim phenomenon is widespread among intensive care nurses in western Austria", although it may correspond to reality, is not evidence generated in this study and should not be stated in the conclusions section.
A rewriting of this section is suggested.
Author Response
Dear Reviewer,
on behalf of all authors I would like to thank you for your valuable comments that helped significantly in improving our paper. We would like to adress your comments point by point as followed.
Kind regards
Reinhard Strametz
The manuscript presented is clear, concise and structured. The research question is explicitly stated and well delimited. The design, methods and instruments used are appropriate and well justified. The qualitative content analysis is appropriate and the results are presented in an orderly manner, answering the research questions. The use of tables is appropriate and well-adjusted. The discussion is presented in an orderly and well-integrated manner. However, the conclusions are vague.
Thank you very much for your overall feedback
For these reasons, a minor revision of the present version is recommended before publication.
Specific remarks:
1- Introduction:
Although the concept of second victim is clearly defined, it encompasses several characteristics. It is suggested to include the specific characteristics or criteria used by the researchers to consider the nursing professional as a second victim and therefore a suitable subject as an informant. I also consider it useful to include in the introduction the concept of patient safety mentioned later in the discussion.
We have added additional information as marked:
Due to the prevailing COVID19 pandemic, the associated exceptional situation in health institutions can be viewed as an unforeseen incident, and the resulting traumatization and stress can lead to the treating practitioners becoming second victims [3, 4]. In addition staff is overloaded and stressed in every aspect due to the scarcity of resources in the health care system resulting in an increased risk for medical errors [5].
...
Second victim traumatization may have negative impact on health worker wellbeing due to dysfunctional coping strategies like self-isolation, reduced self-confidence, flash backs and increased fear of conducting further mistakes [7]. As psychological precursors clearly are one source of increased risk for medical errors [10] it also very likely to affect patient safety of future patients treated after second victim traumatization.
2. Methods:
The methodological design is appropriate and conforms to international COREQ recommendations. The approach, as a phenomenological study, responds to the object of study. The techniques used in the data collection are in accordance with the type of design and are clearly described, as well as the sampling technique (snowball).
As a suggestion for improvement, the question arises as to why there are only 20 individuals, and supposing that it responds to criteria of redundancy of information, it is suggested that this question be explicitly reflected in the manuscript.
We have explicitly stated the reason for limitation of number of interviews as followed:
A total of 20 intensive care nurses from Tyrol and Vorarlberg were interviewed. Sample size was set to this limit to address expected redundancy of information.
3. Results:
Table 2 refers to Symptoms after an adverse event. It reflects psychological and physical symptoms such as internal unrest and drop of quality of life. These symptoms can be interpreted in different ways and are ambiguous, so it is suggested to include the meaning used in the text.
We have clarified symptoms used in table 2 as followed:
The most common symptoms of respondents were feelings of guilt, problems falling or staying asleep, flashbacks of what they had experienced and problems in everyday work such as reduced performance and problems with performing routine activities. Self-doubting themselves also contributed to said problems. In their private life the participants experienced a drop of life quality and internal unrest, as in feeling queasy.
4. Discussion:
Delete the opening sentence : A total of 20 interviews took place in Tyrol and Vorarlberg. It is already included in the results section.
Opening sentence was deleted.
5. Conclusion:
The conclusions are vague and do not give a clear description of the meaning of the phenomenon studied. The phenomenological design selected in the study does not allow, nor is it intended to allow, for generalisation. The sentence "The second victim phenomenon is widespread among intensive care nurses in western Austria", although it may correspond to reality, is not evidence generated in this study and should not be stated in the conclusions section.
A rewriting of this section is suggested.
We rewrote our conclusion. It now reads:
The second victim phenomenon is present among intensive care nurses in western Austria and therefore requires further research. As expressed phenomenons and coping strategies can be compared to previous research in different settings both durther quantitative surveys of intensive care nurses as well as tailored intervention programs addressing prevention and systematic support of second victims should be considered for this target group.